# Dependence of the Physical Properties and Molecular Dynamics of Thermotropic Liquid Crystalline Copolyesters on *p*-Hydroxybenzoic Acid Content

**DOI:** 10.3390/polym12010198

**Published:** 2020-01-12

**Authors:** Gi Tae Park, Won Jun Lee, Jin-Hae Chang, Ae Ran Lim

**Affiliations:** 1Department of Polymer Science and Engineering, Kumoh National Institute of Technology, Gumi 39177, Korea; pkgt0129@naver.com (G.T.P.); wns2536@naver.com (W.J.L.); 2Analytical Laboratory of Advanced Ferroelectric Crystals, Jeonju University, Jeonju 55069, Korea; 3Department of Science Education, Jeonju University, Jeonju 55069, Korea

**Keywords:** thermotropic liquid crystalline copolymer, *p*-hydroxybenzoic acid, thermal property, ^13^C solid-state nuclear magnetic resonance spectroscopy, molecular dynamics

## Abstract

Two series of thermotropic liquid crystal copolymers (TLCPs) with different monomer structures and compositions were synthesized. The copolymers in the first series consisted of 2,5-diethoxyterephthalic acid (ETA), hydroquinone (HQ), and *p*-hydroxybenzoic acid (HBA), whereas those in the second series contained ETA, 2,7-dihydroxynaphthalene (DHN), and HBA. In both series, the molar ratio of HBA to the other monomers varied from 0 to 5. The thermal properties, degree of crystallinity, and stability of the liquid crystalline mesophase of the copolymers obtained at each HBA ratio were evaluated and compared. Overall, at each HBA content, the DHN-containing copolymer had better thermal properties, but the HQ-containing copolymer exhibited a higher degree of crystallinity and a more stable liquid crystalline mesophase. Furthermore, similar thermal stabilities were observed in both series. The dependence of the molecular dynamics of the TLCPs on the monomer structure was explained using ^13^C magic-angle spinning/cross-polarization nuclear magnetic resonance spectroscopy. An in-depth investigation of the relaxation time of each carbon revealed that the molecular motions of the TLCPs were greatly influenced by the structures of the monomers present in the main chain. The molecular dynamics of the HQ and DHN monomers in the two series were evaluated and compared.

## 1. Introduction

Thermotropic liquid crystal polymers (TLCPs), which contain a special monomer chemical structure, are already widely used as high-performance commercial engineering polymers owing to their good thermal properties, high strength, high modulus, low viscosity, and other excellent mechanical properties [1,2]. Many studies have correlated the structures of TLCPs with their characteristics [3,4,5]. In particular, compared with liquid crystalline polyamides, which exhibit lyotropic properties, heat-releasing liquid crystalline polyesters have attracted much attention because of their ease of melting, despite their inferior physical strength [6,7].

Although interest in TLCPs and their composites has been increasing, to achieve enhanced physical properties, most studies have focused on rigid rod-type TLCPs with completely aromatic monomers in the main chain [8,9]. In almost all TLCPs, the thermo-mechanical properties have been enhanced by using straight, rigid rod-shaped monomer units, including terephthalic acid (TPA), hydroquinone (HQ), 4,4′-biphenol (BP), *p*-hydroxybenzoic acid (HBA), 6-hydroxy-2-naphthoic acid (HNA), naphthalenediol derivatives, and naphthalenedicarboxylic acid isomers [9,10,11,12,13]. If the basic structure is substituted with a *para*-substituted carboxyl group or benzene ring, homopolyesters synthesized from monomers such as TPA, HQ, and HBA typically melt at approximately 600 °C [12]. Although rigid rod-type TLCPs exhibit excellent thermal and mechanical properties, their high melting points generally make processing difficult [6,14]. These TLCPs also show low solubilities in common solvents. As a result, many studies have investigated the introduction of substituents, flexible alkyl groups, or side-group TLCPs [15,16,17]. The use of a monomer with a flexible alkyl unit or kinked monomer structure in the main chain or a bulky substituent in the aromatic ring can decrease the melting point of the TLCP to 400 °C or lower, which makes melting or injection molding possible [4,18]. In addition, the melting point can be greatly reduced by using asymmetric monomers or by copolymerizing monomers with different structures such as HBA and HNA [19,20]. If the copolymer is synthesized using a well-designed monomer, it can be easily processed, and lowering the processing temperature can expand its applicability. For example, a structure with a flexible alkyl moieties as side groups or *meta*-substituted monomer units can provide significantly increased processability. Disadvantageously, this improvement in processability is often accompanied by a significant deterioration of the thermo-mechanical properties.

The measurement of ^13^C nuclear magnetic resonance (NMR) chemical shifts is the best method to identify synthesized chemical structures. In addition, ^13^C-NMR relaxation times in the rotating frame spin-lattice (*T_1ρ_*) for ^13^C nuclei, which have low natural abundance, are affected by dipolar interactions with directly bonded hydrogens [21]. Therefore, the determination of the relaxation values of nuclei in different environments in the chemical structure can provide information on the molecular motion occurring in each part of the structure. The ^13^C-NMR *T_1ρ_* values are particularly informative because they are directly related to the motion of each carbon in the kHz frequency range. ^13^C-NMR *T_1ρ_* results obtained using cross-polarization (CP) and magic-angle spinning (MAS) have been used to elucidate the molecular dynamics of various chemical structures [22,23]. Therefore, the determination of ^13^C-NMR *T_1ρ_* values is the best method for studying the molecular dynamics of individual ^13^C atoms in a chemical structure.

The objectives of this study were to: (1) synthesize a two copolymer series using 2,5-diethoxyterephthalic acid (ETA), HBA, and two different diol monomers, namely, HQ and 2,7-dihydroxynaphthalene (DHN); (2) study the effect of the HBA unit on the properties of the ETA/HQ and ETA/DHN copolymers by varying the HBA molar ratio (0–5); and (3) study the molecular dynamics of the two copolymer series using ^13^C solid-state NMR spectroscopy.

The copolymers were synthesized using ETA, which contains dialkoxy side groups, and DHN, which is a kinked monomer, to lower the values the glass transition temperature (*T_g_*), melt transition temperature (*T_m_*), and isotropization temperature (*T_i_*). Both ETA and DHN have very poor thermal properties, and to compensate for this disadvantage, *para*-substituted HBA was also used as a monomer. The thermal properties, thermal stability, liquid crystalline mesophase, degree of crystallinity, and molecular dynamics of the synthesized copolymers were investigated while varying the molar ratio of HBA from 0 to 5. In addition, the effect of the molar ratio of HBA on the structures of the polymers in each series was investigated using ^13^C-NMR spectroscopy. The *T_1ρ_* value for each carbon of the dialkoxy groups, C=O groups, and TPA, HQ, naphthalene, and HBA rings in the two series (TLCP-I and -II) was obtained to understand the molecular dynamics. In addition, the dependence of molecular motions on the molar ratio of HBA was investigated using the relaxation times, and the effect of the molar ratio of HBA on carbon mobility was discussed.

## 2. Materials and Methods 

### 2.1. Materials

All reagents used in this study were purchased from Aldrich Chemical Co. (Yongin, Korea) or TCI (Seoul, Korea) and were used as received. However, common solvents were purified by distillation.

### 2.2. Syntheses of Monomers

The chemical structures of all the monomers (**1**–**5**) for the TLCP syntheses are shown in Figure 1. The monomers were synthesized via several routes [24].

### 2.3. Synthesis of TLCP-I

The TLCPs were synthesized by a melt polymerization method [25]. The monomer composition of the TLCP-I series is shown in Figure 1 and summarized in Table 1, and the detailed reaction conditions are shown in Table 2. The same synthetic procedure was used to produce each TLCP, independent of the monomer composition; therefore, the procedure for synthesizing sample I-C (**2**/**3**/**5** = 1:1:2 (molar ratio)) is detailed here as a representative example. First, 25.42 g (1.0 × 10^−1^ mol) of ETA (**2**), 19.42 g (1.0 × 10^−1^ mol) of 1,4-diacetoxybenzene (**3**), and 36.03 g (2.0 × 10^−1^ mol) of 4-acetoxybenzoic acid (**5**) were placed in a polymerization tube. The mixture was heated under the conditions shown in Table 2 in a constant nitrogen flow. Acetic acid was formed during heating, and the polymerization was completed by lowering the pressure from 300 to 1 Torr as the final step. 

The obtained solid product was cooled to room temperature, washed several times with acetone, and then dried in a vacuum oven at 80 °C for 24 h to obtain TLCP-I. In most common solvents, the synthesized TLCP was not dissolved at all. In particular, it has not been dissolved at all in various mixed solvents, which have often been used to dissolve TLCP. In almost all common solvents, the synthesized TLCP was not dissolved at all. In particular, it did not dissolve at all in mixed solvents such as phenol/p-chlorophenol/1,1,2,2-tetrachloroethane = 25:40:35 (w/w/w), which were frequently used for TLCP dissolution. As no dissolution was observed, the viscosity could not be measured, as indicated in Table 3.

### 2.4. Synthesis of TLCP-II

The same synthetic procedures were used to produce each TLCP; therefore, we describe here the preparation of sample II-C (**2**/**4**/**5** = 1:1:2 (molar ratio)) as a representative example. TLCP-II was synthesized using 25.42 g (1.0 × 10^−1^ mol) of ETA (**2**), 24.42 g (1.0 × 10^−1^ mol) of 2,7-diacetoxynaphthalene (**4**), and 36.03 g (2.0 × 10^−1^ mol) of 4-acetoxybenzoic acid (**5**). As a final step to complete the polymerization, the pressure was lowered from 240 to 1 Torr. The subsequent steps were the same as those described for TLCP-I. The monomer composition of TLCP-II is shown in Figure 1 and summarized in Table 1, and the detailed reaction conditions are shown in Table 2. As described for the TLCP-I series, the solubility of each polymer in the TLCP- II series was examined using a mixture of three solvents. No dissolution was observed; hence, the viscosity was not measured (Table 3).

### 2.5. NMR Spectroscopy

Powdered samples were inserted into 4 mm diameter zirconia rotors and then spun at fsufficient speed to avoid the overlap of spinning sidebands. ^13^C-NMR (Brucker, Berlin, Germany) *T_1ρ_* values were measured by varying the duration of the ^13^C spin-locking pulse [26]. The typical experimental approach assumes the use of CP from protons to enhance ^13^C sensitivity. The width of the π/2 pulse used to measure the ^13^C-NMR *T_1ρ_* values was 3.3 μs. The decay of the ^13^C magnetization in the spin-locking field was followed for spin-locking times of up to 160 ms.

### 2.6. Characterization

The thermal properties of the copolymers were determined by differential scanning calorimetry (DSC), and thermogravimetric analysis (TGA), which were conducted under a N_2_ atmosphere using DuPont 910 equipment (New Castle, DE, USA). The samples were heated or cooled at a rate of 20 °C/min. Wide-angle X-ray diffraction (XRD) measurements were performed at room temperature on a Rigaku (D/Max-IIIB) X-ray diffractometer (Tokyo, Japan) using Ni-filtered Cu-Kα radiation. The scanning rate was 2°/min over a 2*θ* range of 2–35°. A polarizing microscope (Leitz, Ortholux, Lahn-Dill-Kreis, Germany) equipped with a Mettler FP-5 hot stage was used to examine the liquid crystalline behavior. The ChemDradfsw (Bitek Chems. Inc., Seoul, Korea) computer simulation program was used to investigate the three-dimensional (3-D) polymer structures.

The ^13^C NMR chemical shifts and *T_1ρ_* values were obtained by ^13^C CP/MAS NMR spectroscopy at a Larmor frequency of ω_0_/2π = 100.61 MHz using Bruker 400 DSX NMR spectrometers at the Korea Basic Science Institute, Western Seoul Center, Seoul, Korea. The chemical shifts are referenced to tetramethylsilane (TMS).

## 3. Results and Discussion

### 3.1. Thermal Behavior

The results of the 2^nd^ heating were used to obtain thermal properties (*T_g_*, *T_m_*, and *T_i_*) using DSC, and the scanning temperature ranges were determined in advance using TGA to prevent thermal decomposition during scanning. The thermal properties of the two TLCP series are summarized in Table 3. The *T_g_*, which is known to depend on the flexibility and rigidity of the monomers, reflects changes in chain interactions and the free volume. In other words, the *T_g_* is influenced by the segmental motion of the chains and by the substituent size [27]. If the monomer in the main chain has a rigid structure and a large substituent, the free volume will be large, and the *T_g_* value will be high. In the TLCP-I series, the copolymers were synthesized by varying the HBA molar ratio between 0 and 5. The *T_g_* value of the polymer without the HBA was 93 °C, but this value increased to 96 °C when 1 mol of HBA was added. Increasing the HBA amount further to 3 mol decreased the *T_g_* value to 83 °C (sample I-D), but the *T_g_* value then increases to 87 °C when 4 mol of HBA was used. Finally, with 5 mol of HBA, the copolymer maintained a *T_g_* value of 86 °C. 

The *T_g_* values of the TLCP-II series showed a similar trend. When 3 mol of HBA was used (sample II-D), the *T_g_* value was the lowest (99 °C), whereas, at 4 mol, the *T_g_* value increased to 110 °C (sample II-E), which was similar to that obtained with 5 mol of HBA (111 °C). At low ratios of HBA in the random copolymer, the molecular structure was disturbed, resulting in easier chain movement and increased mobility, and consequently, a lower *T_g_*. This phenomenon can be explained thermodynamically using the following equation:*T* = Δ*H*/Δ*S*(1)
where Δ*H* is the enthalpy change and Δ*S* the entropy change. 

However, when the ratio of HBA increases, a block copolymer (poly(hydroxy benzoate) (PHB)) of HBA itself was formed, and the *T_g_* increased. Generally, a block copolymer has a large Δ*H* and a relatively small Δ*S*, resulting in an enhancement of the thermal properties, such as an increase of *T_g_* [28,29]. This phenomenon was observed to be consistent for both series of synthesized TLCPs. The *T_g_* values for the TLCP-II series ranged from 99 to 126 °C, depending on the HBA molar ratio. 

The *T_g_* values of the copolymers containing DHN (TLCP-II series) were higher than those of the polymers containing HQ (TLCP-I series). Compared with the HQ monomer, the naphthalene monomer was substituted with nonlinear units at the 2 and 7 positions. There is a large volume of DHN; therefore, the free volume is large, resulting in higher *T_g_* values for copolymers with DHN than for those with HQ in all HBA molar ranges [19,30]. In other words, the bulky DHN monomer made the movement of the polymer chain difficult, limiting segmental motion, so that the *T_g_* of naphthalene increases more than that of HQ.

The *T_m_* values of the TLCP-I series, which includes HQ, showed a similar tendency to the *T_g_* values (Table 3). The polymer without HBA showed a melting point of 275 °C. However, as the HBA molar ratio in the copolymer increased from 1 to 3, the *T_m_* value decreased from 233 to 228 °C. As the HBA ratio increased, Δ*S* increased, resulting in a decrease in *T_m_* (228 °C). Above 4 mol HBA, the *T_m_* values were in the range of 256–258 °C, as an excess of HBA monomers, resulted in the formation of a block copolymer of HBA, thus increasing the *T_m_* [31]. In the case of the TLCP-II series containing DHN monomers, the copolymer was amorphous, and the *T_m_* was not observed up to an HBA molar ratio of 2 owing to the flexible dialkoxy groups in ETA and the kinked structure of DHN [32]. The flow temperature observed by polarized optical microscopy was approximately 200 °C, regardless of the molar ratio of HBA. However, when the HBA ratio was ≥3, *T_m_* values were observed. On increasing the HBA ratio from 3 to 5, the *T_m_* values gradually increased from 278 to 311 °C. The DSC results for the two TLCP series are shown in Figure 2.

The same tendency was observed for the *T_i_* values as for the *T_g_* and *T_m_* values (Table 3). In the case of the TLCP-I series, the polymer without HBA had a *T_i_* value of 327 °C, but the *T_i_* values of the copolymers with 1–3 mol of HBA monomer content gradually decreased from 321 to 302 °C. However, when the ratio of HBA in the copolymer increased from 3 to 5, the *T_i_* value increased from 302 to 348 °C. In the case of the TLCP-II series, *T_i_* is not observed up to 2 mol of HBA (sample II-C), similar to the behavior observed for *T_m_*. The kinked structure of DHN and the dialkoxy substituents of ETA do not promote liquid crystallinity. In contrast, the simple and linear structure of HBA helps to form liquid crystalline mesophases. Hence, when the molar ratio of HBA increased from 3 to 5, the *T_i_* value increased from 305 to 343 °C. These values were similar to those of TLCP-I (302–348 °C). This phenomenon can also be explained by the rigidity of the HBA monomer in part of the main chain. 

The enthalpy changes of the crystal-anisotropic transition (Δ*H_m_*) and the enthalpy change of the anisotropic-isotropic transition (Δ*H_i_*) were very small, as shown in Table 3, and no constant tendency was found. For example, in the case of TLCP-I, as the number of moles of HBA increased from 0 to 5 moles, Δ*H_m_* were 1.07–2.74 J/g, and Δ*H_i_* were 1.32–3.51 J/g, respectively. In the case of TLCP-II, when the number of moles of HBA increased from 3 to 5 moles, the values of Δ*H_m_* were 2.66–3.42 J/g and Δ*H_i_* was 1.01–2.76 J/g. This result is because the HBA monomer has a random sequence in the melt polymerization process, and the alkoxy side group present in the main chain reduces the effect of enthalpy.

The TGA results for the two TLCP series are shown in Figure 3 and summarized in Table 3. First, in the case of the TLCP-I series, the tendency observed with varying the HBA ratio is similar to the thermal property results (*T_g_*, *T_m_*, and *T_i_*), as described above. A *T_D_^i^* value of 362 °C was observed for HBA = 0 mole (sample I-A), but this value decreased gradually from 380 to 344 °C when HBA was increased from 1 to 3 mol. When HBA was further increased to 5 mol, the *T_D_^i^* value increased again to 356 °C. Similar results were observed for the TLCP-II series, with various changes in *T_D_^i^* depending on the HBA ratio. Therefore, to control the deformation owing to thermal decomposition during processing from the melted state, the structure and ratio of the monomers in the copolymer should be carefully selected. The overall *T_D_^i^* results for both series were generally similar. However, higher *wt_R_^600^* values were observed for the TLCP-II series with DHN monomers than for the TLCP-I series with HQ monomers. This difference is because more charcoal is produced at high temperatures from naphthalene derivatives, which contain two benzene rings, than from HQ, which contains one benzene ring. Overall, the reason why the value of *wt_R_^600^* is generally lower than that of the rigid rod-like main chain TLCP is explained by the alkoxy side group in the main chain.

For the two TLCP series, the changes in the overall thermal properties (*T_g_*, *T_m_*, *T_i_*, and *T_D_^i^*) with the molar ratio of HBA are compared in Figure 4. In each series, the minimum values were obtained when 3 mol of HBA was added to the copolymer. These values then gradually increased as the HBA molar ratio increased up to 5. The shape of the eutectic curve, which depends on the amount of HBA monomer content in the TLCP copolymer, has been described in detail previously, and similar results have been published by several researchers [11,19,32,33]. The temperature change of our study according to the HBA molar ratio is small compared to other research results, which is probably due to the alkoxy side groups in the main chain.

We found that the TLCP copolymer series containing DHN and HQ can be melt-processed without thermal decomposition problems by controlling the ratio of HBA monomer. It was also found that controlling the HBA molar ratio can determine the thermal properties of the TLCP copolymer.

### 3.2. Liquid Crystalline Mesophase

Liquid crystallinity, which occurs between *T_m_* and *T_i_*, can be observed using an optical polarizing microscope [34,35]. Figure 5 shows the liquid crystallinity observed for polymers in the TLCP-I and TLCP-II series at various temperatures. A numer heating and cooling processes were taken to get a better picture, and these LC mesophases were obtained by the heating process between *T_m_* and *T_i_*. All the liquid crystalline mesophases show a thread-like nematic texture [36]. The nematic phases show poorly developed texture, which is mainly due to a high molecular weight or poor flow of the substance above the *T_m_*. 

The stability of the mesophase of an LCP depends on the stiffness and aspect ratio of the mesogenic unit. If the mesogens in the main chain of the polymer are straight and rigid rods, the mesophase of the LCP can be stabilized. Thus, the HBA monomers can stabilize the liquid crystalline phase, regardless of the HBA ratio in the copolymer. However, as in the TLCP-II series, if a 2,7-substituted kinked monomer (DHN) is included in the main chain, the liquid crystalline mesophase is destroyed, and a liquid crystal texture is not observed [37]. As mentioned in the description of *T_m_* and *T_i_*, liquid crystal textures are not observed when the HBA molar ratio is between 0 and 2 in the TLCP-II series.

### 3.3. XRD

The wide-angle XRD patterns of the two TLCP series are shown in Figure 6. Although their diffractograms are different from each other, the XRD patterns of the TLCPs as a whole are not largely different from general crystal characteristics. For the copolymers, major peaks are observed between 2*θ* = 20° and 30°, indicating a semicrystalline character. The degree of crystallinity (DC) was calculated from *Ic*, which is the peak area of the crystalline region, and *Ia*, which is the peak area of the amorphous region, as follows [38]: DC (%) = [*Ic*/(*Ic* + *Ia*)] × 100(2)

The calculated DC values are summarized in Table 3. In the TLCP-I series, the polymer composed of only ETA and HQ had a DC of 39%, whereas the DC of the copolymer with 1 mol of HBA (sample I-B) abruptly decreased to 20%. However, as the molar ratio of HBA in the copolymer increased to 5, DC gradually increased to 39%. As previously mentioned, short and rigid HBAs contribute to the crystallinity of the entire copolymer. Thus, an increase in DC will occur at higher HBA ratios. In contrast, in the TLCP-II series, an amorphous diffraction pattern was observed at an HBA ratio of 1 owing to ETA having flexible alkyl groups and DHN having a kinked structure. However, at an HBA molar ratio of 2 (sample II-C), a very small crystalline peak was observed. When the HBA ratio was increased to 5, the intensity of the peak increased further. As shown in Table 3, when the ratio of HBA in the copolymer increases from 2 to 5, the DC increases from 3% to 18%. Comparing the two TLCP series, it was found that the linear structure of the HQ monomer had a greater effect on the crystallization of the LCP main chain than the kinked structure of the DHN monomer. Based on the results in Figure 6, the *d* and *2θ* values of each XRD peak are summarized in Table 4 [39].

XRD peaks were investigated between *T_m_* and *T_i_* ranges showing LC mesophase, and the results are shown in Figure 7. As expected, the XRD obtained at 285 °C was nearly amorphous, and the sharp peaks were almost absent compared to the results obtained at 25 °C. In high-temperature conditions, structural irregularity caused by a random sequence of monomer units, together with the random existence of alkoxy side group or kinked structures, certainly would hinder crystallization. 

Figure 8 shows a 3-D computer simulation illustrating the detailed relationship between the copolymer structure and crystallinity. The chemical structures of copolymers in the two series obtained using the same molar ratios of HBA (1 and 5 mol HBA) are compared. In the TLCP-I series, the copolymer obtained using 5 mol of HBA (sample I-F) shows a more linear structure than that obtained using 1 mol of HBA (sample I-B). This result is due to the effect of the rigid rod-shaped HBA monomer on the crystallinity.

Similar results were obtained for the TLCP-II series, as the copolymer containing 5 mol of HBA shows a more linear structure than that containing 1 mol of HBA (sample II-B), which is expected to affect the crystallinity. In contrast, when comparing the 3-D structures of the two TLCP series at the same HBA ratio, the TLCP-I structure is more linear compared to the TLCP-II structure, but the structure of the TLCP-II is more spherical. Thus, it was found that the copolymer structures directly affect the crystallinity.

### 3.4. ^13^C Chemical Shifts and Relaxation Times 

Structural analysis of the TLCP-I and TLCP–II series was carried out by solid-state ^13^C CP/MAS NMR. The ^13^C chemical shifts of the TLCP-I and -II series were obtained for the carbons of the alkoxy groups and aromatic rings at room temperature. In the TLCP-I series, for sample I-A, the ^13^C chemical shifts for CH_3_ and CH_2_ of the alkoxy group are observed at 14.79 and 64.69 ppm, respectively, as shown in Figure 9. The peaks at 115.96, 122.64, 148.87, and 151.63 ppm are assigned to the benzene rings in ETA and HQ, and the chemical shift for C=O is observed at 164.78 ppm [40,41]. The peak for the carbon of C=O bonds has a relatively low intensity. The spinning sidebands for the benzene rings in ETA and HQ are marked with asterisks in Figure 9. The chemical shifts of all carbons are consistent with the chemical structure shown in Figure 9. In the case of sample I-F (Figure 10), the ^13^C chemical shifts for the alkoxy group, ETA, HQ, and C=O are similar to those observed in I-A. The ^13^C chemical shifts for the benzene ring in PHB are located at 126.37, 132.34, and 148.70 ppm. 

By contrast, in the TLCP-II series, for sample II-A, the ^13^C chemical shifts of CH_3_ and CH_2_ are observed at 14.36 and 64.60 ppm, respectively (Figure 11). The ^13^C peak at 164.11 ppm corresponds to C=O, and the signals at 149.53, 133.98, 128.91, and 119.19 ppm are attributed to the aromatic rings in ETA and DHN [42]. The asterisks in Figure 11 represent the spinning sidebands of ETA and DHN rings. The ^13^C chemical shifts for the alkoxy group, C=O, and the aromatic rings in ETA, DHN, and PHB in sample II-F are consistent with the chemical structure shown in Figure 12. The results for II-F are similar to those for samples II-B, II-C, II-D, and II-E. 

To obtain the ^13^C-NMR *T_1ρ_* values, the magnetization recovery curves for the TLCP-I and II series were measured as a function of the delay time. All the magnetization recovery traces can be described by a single-exponential function [43]:*I*(*t*) = *I_0_* exp(‒*Wt*),(3)
where *I*(*t*) is the magnetization according to the spin-locking pulse duration *t*, and *I*_0_ is the total nuclear magnetization at thermal equilibrium. The *T*_1*ρ*_ (=1/*W*) values were obtained from the slopes of the delay time vs. intensity curves, and the results for each carbon in the TLCP-I and -II series are listed in Table 5 and Table 6. The ^13^C relaxation times in the TLCP-I and -II series are compared according to the molar ratio of HBA. The ^13^C-NMR *T_1ρ_* values for the CH_3_ and CH_2_ of sample I-A are 39.2 and 4.4 ms, respectively (Table 5). The CH_3_ group has a longer relaxation time than the CH_2_ of the alkoxy groups, which is consistent with the fact that dipolar relaxation is more efficient based on the number of bonded protons. In addition, the carbonyl carbons (C=O) have greater *T_1ρ_* values than the carbons of the alkoxy group. When the molar ratio of HBA is increased, the ^13^C-NMR *T_1ρ_* values for the alkoxy group, C=O, and the aromatic rings in TPA, HQ, and HBA decrease, as shown in Table 5.

The ^13^C-NMR *T_1ρ_* values for the TLCP-II series are shown in Table 6. Here, the ^13^C peak for the benzene ring in HBA somewhat overlaps the peak for the aromatic rings of ETA and HQ, as shown in Figure 12, making it difficult to obtain the *T_1ρ_* values. For C=O in sample II-A, the *T_1ρ_* value is remarkably high. Thus, the alkoxy group and C=O in the TLCP-II series have higher mobilities than those in the TLCP-I series. Carbons e and f in sample I-A and carbons g, h, and i in sample II-A have longer relaxation times than the other carbons, as shown in Table 5 and Table 6. These greater *T_1ρ_* values indicate a higher rigidity of the main chains, which implies that there are strong interfacial interactions between the aromatic rings and the polymer main chains and between the alkoxy groups and polymer main chains. The *T_1ρ_* values for the TLCP-I and -II series indicate the effect of the HBA molar ratio on the mobility. All of the C atoms in the TLCP-II series have higher mobilities than those in the TLCP-I series.

## 4. Conclusions

Copolymers in the TLCP-I series included ETA, HQ, and HBA, whereas those in the TLCP-II series included ETA, DHN, and HBA. Both series were synthesized with varying molar ratios between HBA and the other two monomers (0–5). The thermal properties, liquid crystalline mesophases, and degrees of crystallinity of the polymers in the two TLCP series were investigated and compared. In both the TLCP series, the minimum *T_g_*, *T_m_*, *T_i_*, and *T_D_^i^* values were observed at an HBA molar ratio of 3. The liquid crystalline mesophases showed nematic schlieren textures in both series, but for the TLCP-II series, nematic textures were not observed at HBA ratios lower than 3 owing to the kinked DHN structure in the main chain. The degrees of crystallinity of the TLCP-I series containing the linear HQ monomer were higher than those of the TLCP-II series with the kinked DHN monomer.

The chemical structures of the polymers in the TLCP-I and TLCP-II series were confirmed based on the ^13^C chemical shifts. From the *T_1ρ_* values, the effects of the HBA molar ratio on carbon mobility were determined for the TLCP-I and -II series. The ^13^C-NMR *T_1ρ_* values decreased for all carbon atoms as the molar ratio of HBA monomers bound to the HQ, and DHN rings increased. In the TLCP-I series, the *T_1ρ_* values decreased gradually with an increase in the HBA molar ratio, whereas in the TLCP-II series, the *T_1ρ_* values decreased more sharply. This difference indicates that the polymers in the TLCP-II series are more rigid than those in the TLCP-I series and that the molar ratio of HBA has a greater influence on the TLCP-II series. Thus, the HBA rings in the TLCP-I series have higher mobility than those in the TLCP-II series. The ^13^C-NMR *T_1ρ_* was dominated by fluctuations in the anisotropic chemical shifts and became shorter when the amplitude of the molecular motions decreased.

## Figures and Tables

**Figure 1 polymers-12-00198-f001:**
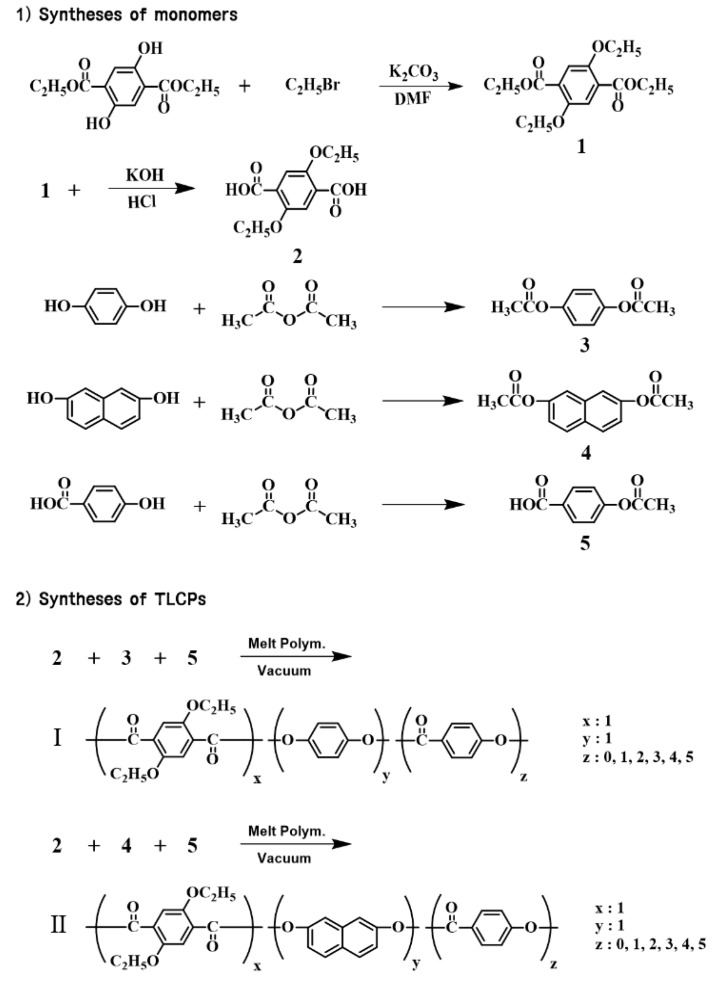
Synthetic routes of the thermotropic liquid crystalline polymers.

**Figure 2 polymers-12-00198-f002:**
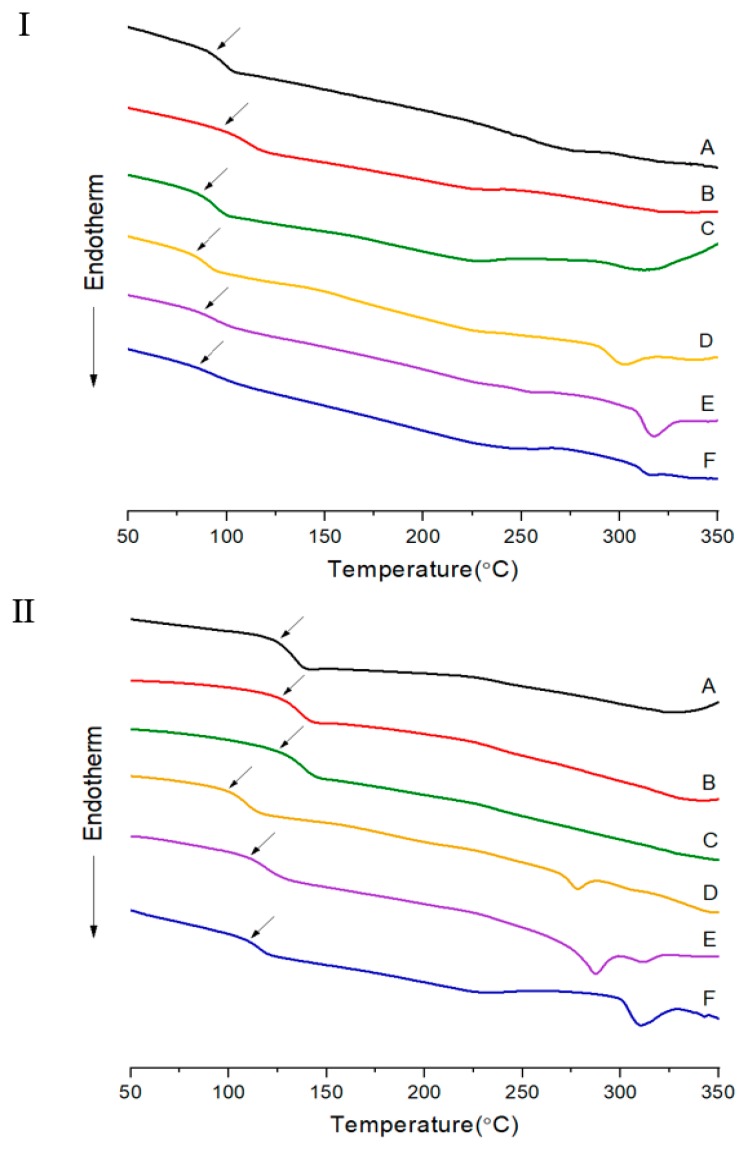
DSC thermograms of TLCP-I and TLCP-II series.

**Figure 3 polymers-12-00198-f003:**
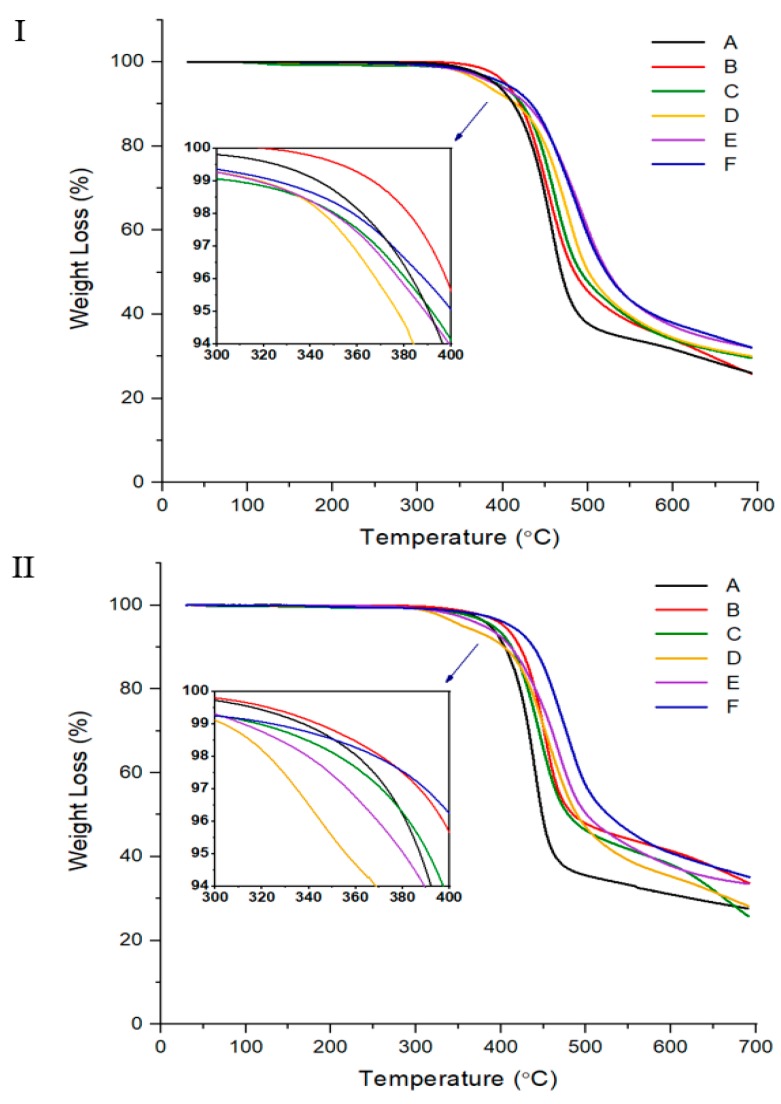
TGA thermograms of TLCP-I and TLCP-II series.

**Figure 4 polymers-12-00198-f004:**
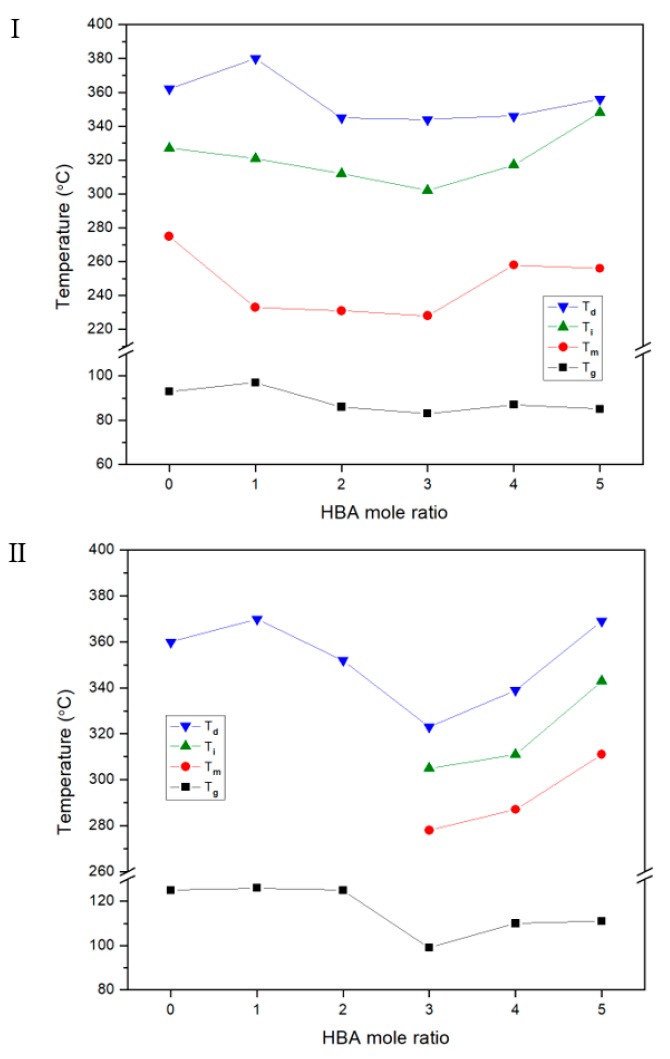
Thermal properties of TLCP-I and TLCP-II series with various HBA ratios.

**Figure 5 polymers-12-00198-f005:**
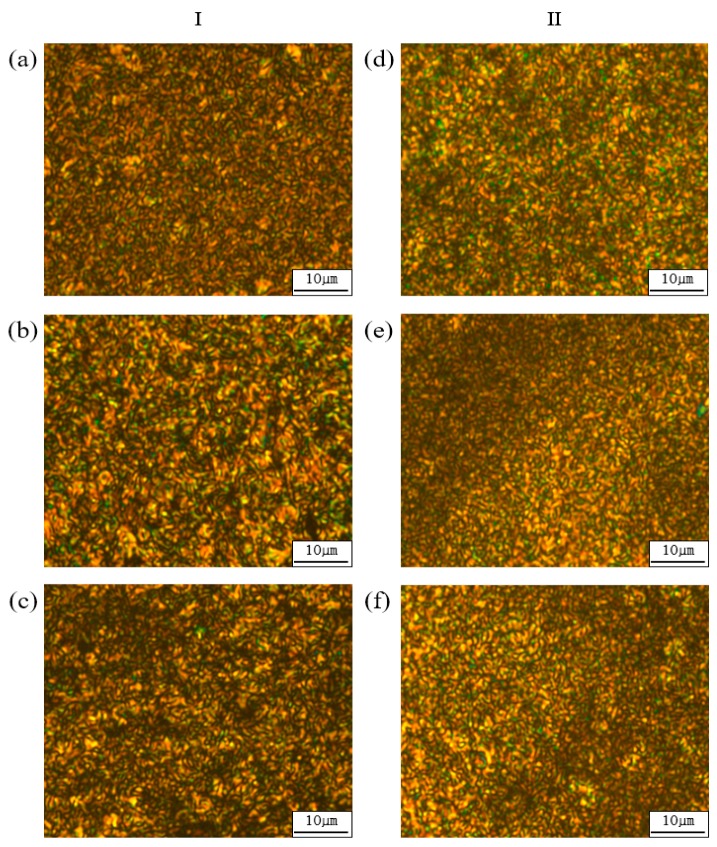
Polarized optical micrographs of (**a**) I-D at 280 °C, (**b**) I-E at 300 °C, (**c**) I-F at 270 °C, (**d**) II-D at 280 °C, (**e**) II-E at 295 °C, and (**f**) II-F at 315 °C (magnification 200×).

**Figure 6 polymers-12-00198-f006:**
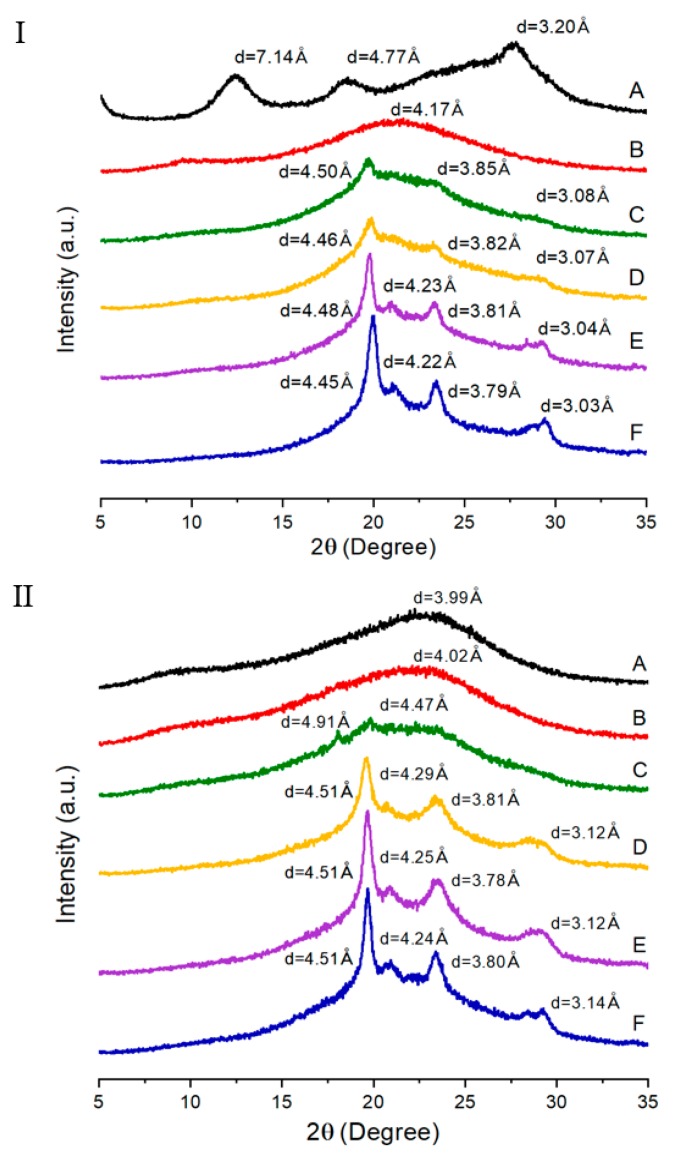
XRD patterns of TLCP-I and TLCP-II series.

**Figure 7 polymers-12-00198-f007:**
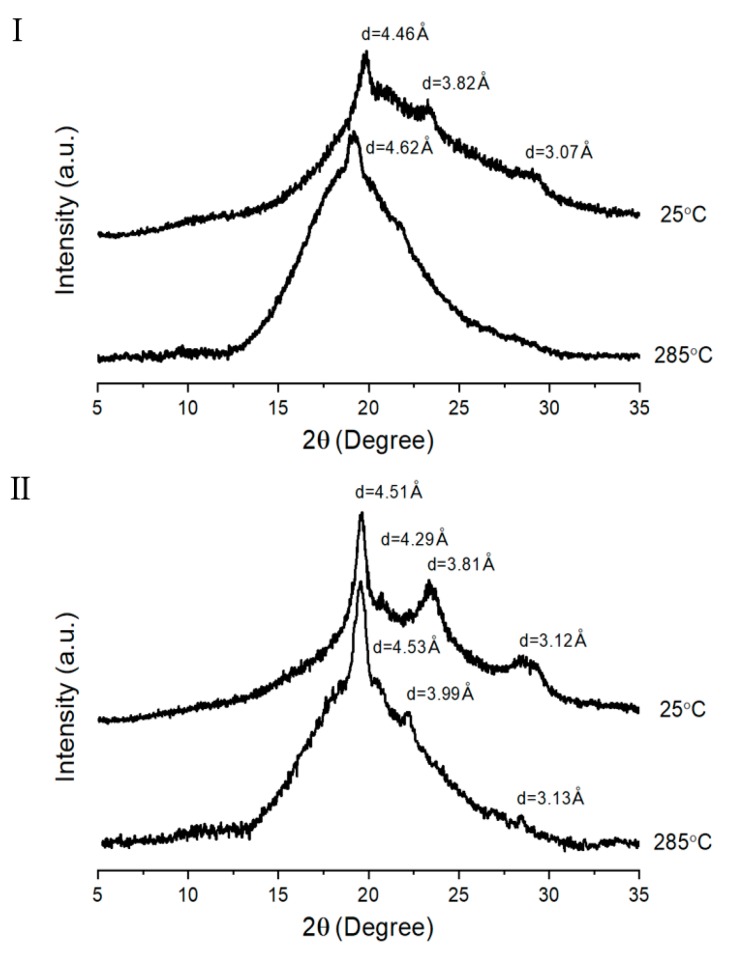
Patterns of sample D in two TLCP series at 25 °C and 285 °C.

**Figure 8 polymers-12-00198-f008:**
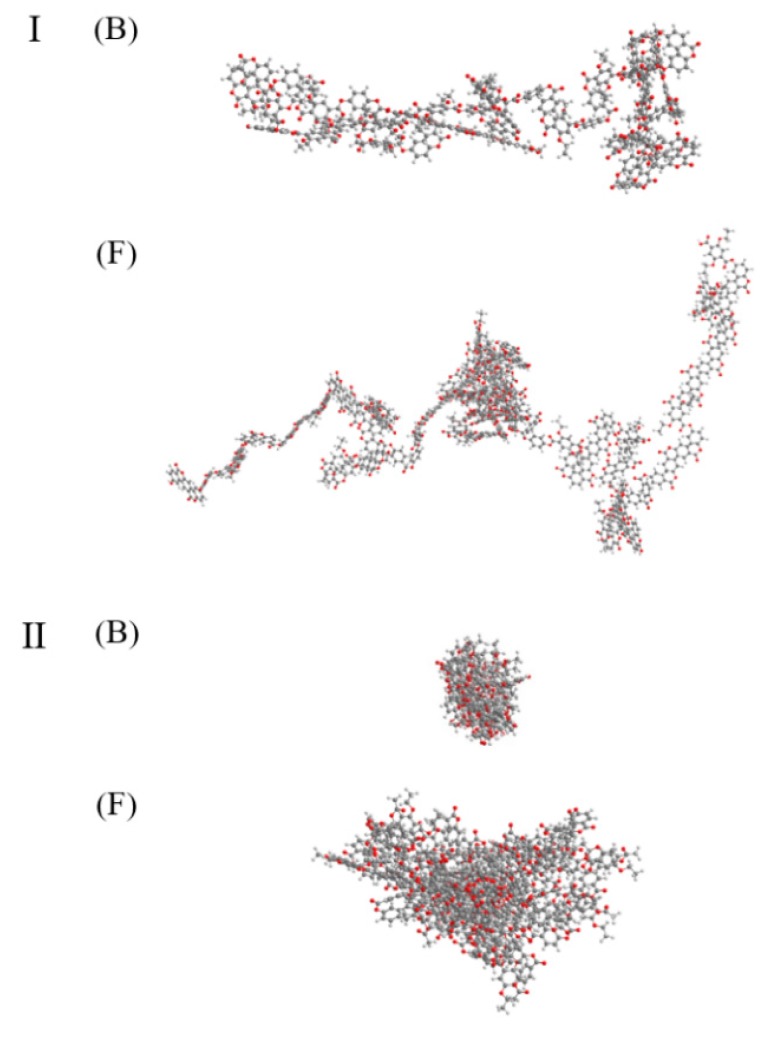
Comparison of the 3-D chemical structures of samples B and F in two TLCP series.

**Figure 9 polymers-12-00198-f009:**
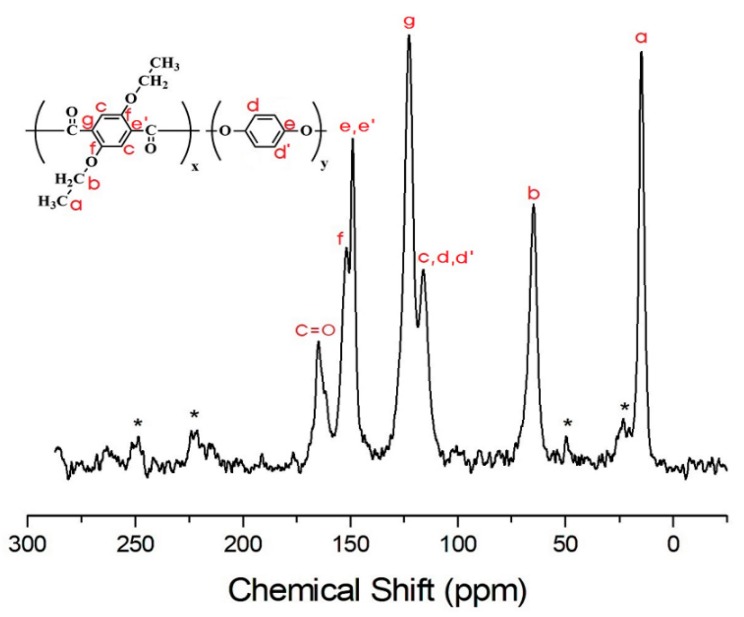
^13^C-NMR chemical shifts of sample I-A at room temperature. In the chemical formula, x:y = 1:1. The spinning sidebands are marked with asterisks.

**Figure 10 polymers-12-00198-f010:**
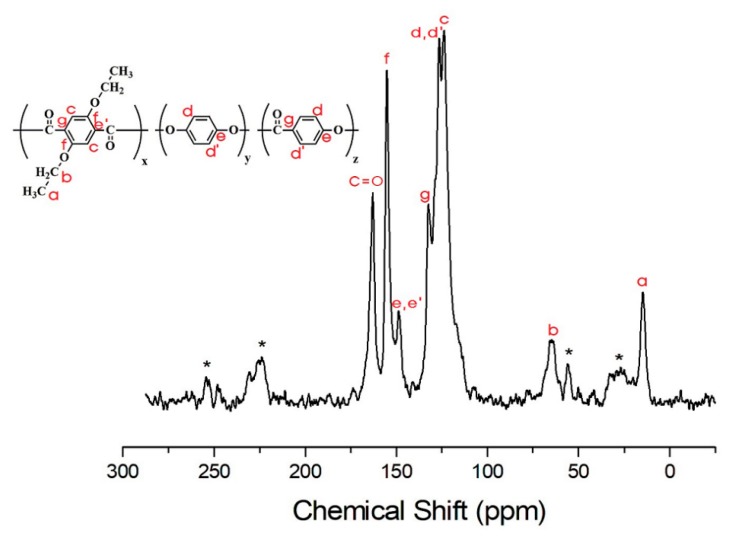
^13^C-NMR chemical shifts of sample I-F at room temperature. In the chemical formula, x:y:z = 1:1:5. The spinning sidebands are marked with asterisks.

**Figure 11 polymers-12-00198-f011:**
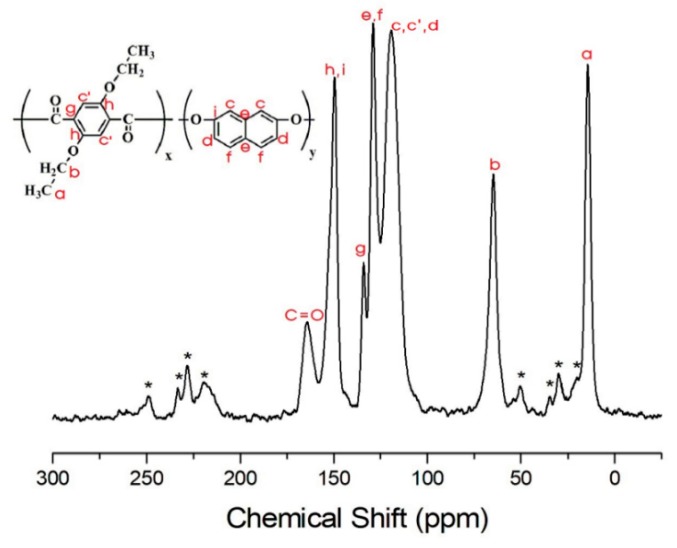
^13^C-NMR chemical shifts of sample II-A at room temperature. In the chemical formula, x:y = 1:1. The spinning sidebands are marked with asterisks.

**Figure 12 polymers-12-00198-f012:**
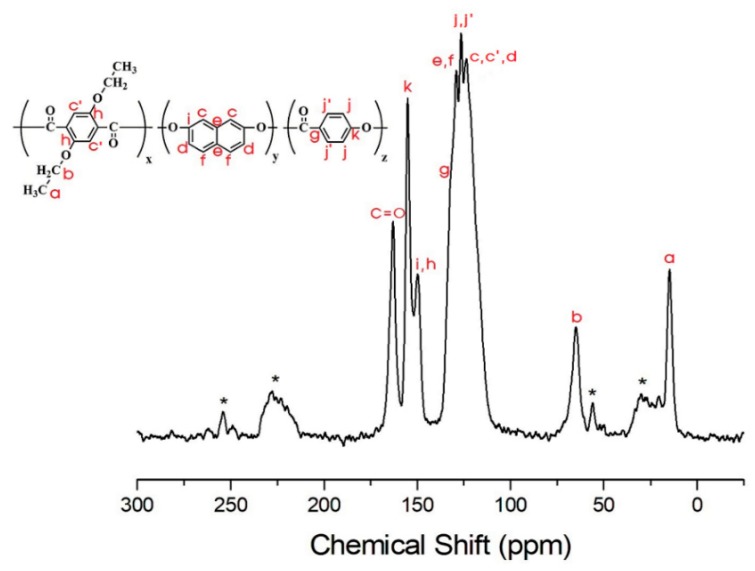
^13^C-NMR chemical shifts of sample II-F at room temperature. In the chemical formula, x:y:z = 1:1:5. The spinning sidebands are marked with asterisks.

**Table 1 polymers-12-00198-t001:** Molar ratios of monomers in the TLCPs.

TLCP	I ^a^			II ^b^		
	ETA	HQ	HBA	ETA	DHN	HBA
A	1	1	0	1	1	0
B	1	1	1	1	1	1
C	1	1	2	1	1	2
D	1	1	3	1	1	3
E	1	1	4	1	1	4
F	1	1	5	1	1	5

^a^ x: 2,5-diethoxyterephthalic acid (ETA), y: hydroquinone (HQ), z: *p*-hydroxybenzoic acid (HBA). ^b^ x: ETA, y: 2,7-dihydroxynaphthalene (DHN), z: HBA.

**Table 2 polymers-12-00198-t002:** Melt polymerization conditions of TLCPs.

TLCP	I ^a^	II ^b^
	Temperature (°C)/time (min)/pressure (Torr)	Temperature (°C)/time (min)/pressure (Torr)
A	240/120/760 → 260/120/760 → 280/60/760 → 290/60/760 → 300/40/760 → 310/40/300 → 320/30/1	250/60/760 → 270/50/760 → 280/30/760 → 290/50/760 → 290/30/240 → 290/50/1
B	240/120/760 → 260/80/760 → 280/60/760 → 300/45/760 → 300/30/300 → 310/60/1	250/40/760 → 270/120/760 → 270/60/240 → 270/40/1
C	240/150/760 → 260/60/760 → 280/90/760 → 300/75/760 → 310/35/300 → 310/40/1	250/30/760 → 270/30/760 → 280/30/760 → 290/80/760 → 290/30/240 → 290/50/1
D	240/120/760 → 260/60/760 → 280/60/760 → 300/60/760 → 305/45/300 → 310/30/1	245/120/760 → 250/30/760 → 265/50/760 → 270/50/240 → 285/40/1
E	240/120/760 → 260/60/760 → 280/60/760 → 300/60/760 → 305/35/300 → 310/40/1	250/120/760 → 265/50/760 → 280/60/760 → 285/50/240 → 290/30/1
F	245/120/760 → 260/60/760 → 280/60/760 → 300/60/760 → 310/30/300 → 310/30/1	260/60/760 → 270/60/760 → 280/60/760 → 295/30/760 → 295/30/240 → 295/30/1

^a^ x: 2,5-diethoxyterephthalic acid (ETA), y: hydroquinone (HQ), z: *p*-hydroxybenzoic acid (HBA). ^b^ x: ETA, y: 2,7-dihydroxynaphthalene (DHN), z: HBA.

**Table 3 polymers-12-00198-t003:** General properties of TLCPs.

TLCP	Ⅰ ^a^										Ⅱ ^b^										
	IV ^c^	*T_g_* (°C)	*T_m_* (°C)	*T_i_* (°C)	*∆H*_m_ (J/g)	*∆H_i_* (J/g)	*T_D_*^i d^ (°C)	wt_R_^600 e^ (%)	LC Phase	DC ^f^ (%)	IV	*T_g_* (°C)	*T_f_*^g^ (°C)	*T_m_* (°C)	*T_i_* (°C)	*∆H*_m_ (J/g)	*∆H_i_* (J/g)	*T_D_*^i^ (°C)	wt_R_^600^ (%)	LC Phase	DC (%)
A	Insol.^h^	93	275	327	1.07	1.32	362	32	Nematic	39	Insol.	125	200					360	31	No.	0
B	Insol.	96	233	321	1.28	1.42	380	34	Nematic	20	Insol.	126	200					370	41	No.	0
C	Insol.	86	231	312	1.69	2.07	345	34	Nematic	20	Insol.	125	200					352	38	No.	3
D	Insol.	83	228	302	1.61	2.50	344	34	Nematic	24	Insol.	99		278	305	2.66	1.04	323	36	Nematic	15
E	Insol.	87	258	317	1.61	3.51	346	37	Nematic	26	Insol.	110		287	311	3.01	2.76	339	38	Nematic	16
F	Insol.	86	256	348	2.74	1.47	356	38	Nematic	39	Insol.	111		311	343	3.42	1.01	369	41	Nematic	18

^a^ x: 2,5-diethoxyterephthalic acid (ETA), y: Hydroquinone (HQ), z: *p*-hydroxybenzoic acid (HBA). ^b^ x: ETA, y: 2,7-dihydroxynaphthalene (DHN), z: HBA. ^c^ Inherent viscosity was measured at a concentration of 0.1 g/dL solution in phenol/p-chlorophenol/TCE = 25/40/35 (w/w/w) at 25 °C. ^d^ At 2% initial weight-loss temperature. ^e^ Weight percent of residue at 600 °C. ^f^ Degree of crystallinity. ^g^ Flow temperature is observed by polarized optical micrographs. ^h^ Insoluble.

**Table 4 polymers-12-00198-t004:** *d* values corresponding to the XRD peaks of the TLCPs.

TLCP	*d* (Å) (2*θ* (degree))							
	I ^a^				II ^b^			
A	7.14 (12.38) ^c^	4.77 (18.56)	3.20 (27.82)	-	3.99 (22.24)	-	-	-
B	4.17 (21.28)	-	-	-	4.02 (22.1)	-	-	-
C	4.50 (19.72)	3.85 (23.08)	3.08 (28.96)	-	4.91 (18.04)	4.47 (19.84)	-	-
D	4.46 (19.88)	3.82 (23.26)	3.07 (29.06)	-	4.51 (19.64)	4.29 (20.7)	3.81 (23.34)	3.12 (28.56)
E	4.48 (19.78)	4.23 (20.96)	3.81 (23.32)	3.04 (29.3)	4.51 (19.68)	4.25 (20.88)	3.78 (23.52)	3.12 (28.58)
F	4.45 (19.94)	4.22 (21.02)	3.79 (23.44)	3.03 (29.4)	4.51 (19.68)	4.24 (20.94)	3.80 (23.4)	3.14 (28.42)

^a^ x: 2,5-diethoxyterephthalic acid (ETA), y: hydroquinone (HQ), z: *p*-hydroxybenzoic acid (HBA). ^b^ x: ETA, y: 2,7-dihydroxynaphthalene (DHN), z: HBA. ^c^ 2*θ* values are shown in parentheses.

**Table 5 polymers-12-00198-t005:** Spin-lattice relaxation time *T_1ρ_* (ms) in the rotating frame for each carbon in the TLCP-I series at room temperature.

TLCP	I-A	I-B	I-C	I-D	I-E	I-F
CH_3_-a	39.2	39.7	23.8	24.4	25.4	25.7
CH_2_-b	4.4	3.5	4.1	3.4	5.1	2.9
c	12.2	17.4	11.9	14.7	14.1	14.0
d	41.9	46.0	42.9	42.9	47.3	45.2
e	125.9	95.9	90.8	80.4	72.9	66.6
f	121.0	81.5	83.1	70.4	77.8	87.8
g	12.2	8.4	6.7	6.1	5.8	6.7
C=O	92.0	62.2	56.0	59.1	65.8	65.8

**Table 6 polymers-12-00198-t006:** Spin-lattice relaxation time *T_1ρ_* (ms) in the rotating frame for each carbon in the TLCP-II series at room temperature.

TLCP	II-A	II-B	II-C	II-D	II-E	II-F
CH_3_-a	55.1	42.5	38.5	34.0	24.5	36.6
CH_2_-b	8.2	5.8	5.8	6.4	4.5	6.7
c,d	29.3	21.2	18.0	19.0	18.0	27.4
e,f	68.9	53.1	42.7	23.9	22.3	20.5
g	119.9	118.0				
h,i	191.7	144.1	136.5	130.1	106.5	108.1
j				80.5	75.2	
k			107.8	114.2	101.1	94.6
C=O	182.6	101.1	100.4	102.3	83.1	77.6

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
