# Peer review of "Dependence of the Physical Properties and Molecular Dynamics of Thermotropic Liquid Crystalline Copolyesters on p-Hydroxybenzoic Acid Content"

_polymers, 2020, doi:10.3390/polym12010198_

Round 1

Reviewer 1 Report

Manuscript polymers-668240, entitled “Dependence of the Physical Properties and Molecular Dynamics of Thermotropic Liquid Crystalline Copolyesters on p-Hydroxybenzoic Acid Content”, by Gi Tae Park, Won Jun Lee, Jin-Hae Chang and Ae Ran Lim, deals with the synthesis, spectroscopic characterization, thermal properties, liquid crystal behavior and molecular dynamics of two series of copolymers based on the use of differente monomers able to form ester linkages.

The article contains an extensive piece of work and the authors have synthetized a large amount of compounds and used a variety of experimental techniques. However, the manuscript is not appropriate for publication in its present form.

There are a number of points that are not clear:

1- The solubility was examined in a mixture of three particular solvents. Why these solvents? No dissolution was observed in these solvents and therefore the viscosity could not be measured. Why the authors did not test other solvents?

2- The DSC study only covers one heating process. However, for an appropriate calorimetry investigation several scans should be carried out. In particular, the first cooling and second heating are as useful as (or more useful than) the first heating, because they are more representative. If the reason for not carrying out successive scans is the decomposition observed on first heating, the heating should be stopped at a temperature just above isotropization and below decomposition.

3- The effect of varying the percentage of HBA in the different copolyesters on the thermal properties is not clear. Although the authors try to explain the evolution of the glass transition temperatures and the transition temperatures on the basis of the formation of block polymer regions as the percentage of HBA increases, the temperature variations are not significant and it is hard to draw any conclusion. 

4- In figure 2 the Tm transitions are not visible in the DSC curves. Some Ti transitions are not visible either, even for some compounds  in series I. Moreover, in table 3 or figure 2 the transition enthalpies are not given. The enthalpy values are very useful to understand the thermal behavior.

5- The explanation of the differences in the wtr600 values for the two series based on the higher charcoal production for naphthalene derivatives it is difficult to understand. The possible differences in charcoal production are not so high so to account for those differences. Moreover, the observed differences in wtr600 are not very large.

6- In section “Liquid Crystalline Mesophase” it is mentioned that the nematic phases show poorly developed textures. However, in the text and in figure 5 there is no information about if this behavior occurs on heating or on cooling. Very often well-developed textures are only observed on cooling from the isotropic liquid.

7- In the same section, the authors say that rigid and straight monomers in the main chain stabilize the mesophase. It is true that the compounds in series II are not LC at low percentages of HBA. However, for higher percentages the Ti temperatures are similar for both series of compounds.

8- In the XRD section, some small diffraction peaks are not labelled in figure 6 or listed in Table 4. In addition, the compounds should have been studied by X-ray diffraction at high temperatures in the nematic mesophase, and not only in the crystalline phase.

9- The structures of the molecules in Figure 7 are very difficult to see. Moreover, the authors do not say which 3D simulation software has been used to predict the molecule structures. Furthermore the authors propose that the TLCP-I structure is entirely linear. This linearity is not visible in the figure.

10- The 13C-NMR study is rather deficient:

(a) The authors do not explain how they assign the different signals to each carbon atom.

(b) Some carbon atoms are not properly labelled in figures 9, 10 and 11. For example, in figure 9 two different kinds of carbon atom in different rings have been labelled “e”. The same stands for different carbons labelled “d”. They should have been labelled e, e’ and d, d’. The same problem is detected in figure 10 with carbons labelled “c” and in figure 11 with carbons labelled “c” and “j”.

(c) Carbon “g” appears in the spectrum of figure 8 at a shift value very different from that observed in the spectrum of figures 9, 10 and 11.

11- In the conclusions, there are two sentences that apparently are contradictory: on the one hand the authors say that the polymers in the TLCP-II series are more rigid than those in the TLCP-I series and on the other hand they say that that the HBA rings in the TLCP-I series have lower mobility than those in the TLCP-II series.

12- In the last paragraph of the Introduction (line 76): “meta-substituted monomer” should be “bent monomer” or “kinked monomer”.

Author Response

Dear Editor

This is our response to your comments regarding our paper “Dependence of the Physical Properties and Molecular Dynamics of Thermotropic Liquid Crystalline Copolyesters on p-Hydroxybenzoic Acid Content (polymers-668240) in polymers.

Thank you very much for the reviewer's comments. I have carefully revised the manuscript following the comments of the reviewer.

Response to Reviewer-1 comments: Red color

Point-1: The solubility was examined in a specific mixture solvent.

Answer-1: The synthesized TLCP was insoluble in most of the solvents commonly used. A detailed explanation is attached. See page 3, line 111. “In almost all common solvents, the synthesized TLCP was not dissolved at all. In particular, it did not dissolve at all in mixed solvents such as phenol/p-chlorophenol/1,1,2,2-tetrachloroethane = 25:40:35 (w/w/w), which were frequently used for TLCP dissolution.”

Point-2: Heat treatment process of DSC scanning.

Answer-2: A detailed explanation is attached. See page 6, line 167: “The results of 2nd heating were used to obtain thermal properties (Tg, Tm, and Ti) using DSC, and the scanning temperature ranges were determined in advance using TGA to prevent thermal decomposition during scanning.”

Point-3. Describe the thermal properties of TLCPs with varying HBA ratios.

Answer-3: As the reviewer pointed out, the thermal properties of the various molar ratios of the HBA were explained and references were supplemented. See page 9, line 254: “The shape of the eutectic curve, which depends on the amount of HBA monomer content in the TLCP copolymer, has been described in detail previously, and similar results have been published by several researchers [11,19,32,33]. The temperature change of our study according to HBA molar ratio is small compared to other research results, which is probably due to the alkoxy side groups in the main chain.”

Point-4: Supplement the description of enthalpy value.

Answer-4: As the reviewer pointed out, the explanation for the enthalpy value was supplemented.

See page 8, line 226: “The enthalpy changes of the crystal-anisotropic transition (ΔHm) and the enthalpy change of the anisotropic-isotropic transition (ΔHi) were very small as shown in Table 3, and no constant tendency was found. For example, in the case of TLCP-I, as the number of moles of HBA increased from 0 to 5 moles, ΔHm were 1.07-2.74 J/g and ΔHi were 1.32-3.51 J/g, respectively. In the case of TLCP-II, when the number of moles of HBA increased from 3 to 5 moles, the values of ΔHm were 2.66-3.42 J/g and ΔHi was 1.01-2.76 J/g. This result is because the HBA monomer has a random sequence in the melt polymerization process and the alkoxy side group present in the main chain reduces the effect of enthalpy.”

In addition, enthalpy values for each DSC curve were supplemented in Table 3.

Point-5: The explanation of the differences in the wtR 600 values for the two series.

Answer-5: This is explained in detail on page 8. See page 8, line 242: “However, higher wtR600 values were observed for the TLCP-II series with DHN monomers than for the TLCP-I series with HQ monomers. This difference is because more charcoal is produced at high temperatures from naphthalene derivatives, which contain two benzene rings, than from HQ, which contains one benzene ring. Overall, the reason why the value of wtR600 is generally lower than that of the rigid rod-like main chain TLCP is explained by the alkoxy side group in the main chain.”

Point-6: Method of heating or cooling to get a picture

Answer-6: As the reviewer pointed out, we explained how to get a better picture. See page 9, line 270: “Several heating and cooling processes were taken to get a better picture and these LC mesophases were obtained by heating process between Tm and Ti.”

Point-7: Why do the two series show similar Ti values at higher HBA concentrations?

Answer-7: A detailed description has been added about Ti. See page 7, line 222: “In contrast, the simple and linear structure of HBA helps to form liquid crystalline mesophases. Hence, when the molar ratio of HBA increased from 3 to 5, the Ti value increased from 305 to 343 °C. These values were similar to those of TLCP-I (302-348 oC). This phenomenon can be also explained by the rigidity of the HBA monomer in part of the main chain.”

Point-8: Specify a small peak value in Figure 6 and show the XRD patterns in the liquid crystallne region.

Answer-8: The small peaks are assigned in Figure 6 and the results are also listed in Table 4. The XRD patterns obtained in the liquid crystalline range are also shown in Figure 7. The explanation for Figure 7 is also supplemented.

See page 12, line 321: “XRD peaks were investigated between Tm and Ti ranges showing LC mesophase and the results are shown in Figure 7. As expected, the XRD obtained at 285 oC was nearly amorphous, and the sharp peaks were almost absent compared to the results obtained at 25 oC. In high temperature conditions, structural irregularity caused by a random sequence of monomer units, together with the random existence of alkoxy side group or kinked structures, certainly would hinder crystallization.”

Point-9: Product description of the 3D simulation software and supplementary explanation of the linearity shown in Figure 8,

Answer-9: 3D simulation software is available in Bitek Chems. Inc. products. The explanation of linearity shown in Figure 8 is described on page 13, line 340: “In contrast, when comparing the 3-D structures of the two TLCP series at the same HBA ratio, the TLCP-I structure is more linear compared to the TLCP-II structure, but the structure of the TLCP-II is more spherical. Thus, it was found that the copolymer structures directly affect the crystallinity.”

Point-10-(a): Describe how each carbon is assigned

Answer-10-(a): Each carbon position was assigned by reference to the C-13 NMR book. Reference 41 has been supplemented in the manuscript.

Point-10-(b): Not properly labelled in Figures.

Answer-10-(b): As the reviewer pointed out, the label of the carbon element was modified in each Figure. See Figures 9-12.

Point-10-(c): About carbon “g”.

Answer-10-(c): Corrected the position of "g" in the figure and supplemented on line 351, page 13. “The peaks at 115.96, 122.64, 148.87, and 151.63 ppm are assigned to the benzene rings in ETA and HQ,”

Point-11: Contradictions in describing the mobility of the two TLCP series

Answer-11: As the reviewer pointed out, the mobility of the two TLCP series is discussed in the conclusion. See page 17, line 431. “This difference indicates that the polymers in the TLCP-II series are more rigid than those in the TLCP-I series, and that the molar ratio of HBA has a greater influence on the TLCP-II series. Thus, the HBA rings in the TLCP-I series have higher mobility than those in the TLCP-II series.”

Point-12: Meta-substituted monomer should be bent monomer or kinked monomer

Answer-12: As the reviewer pointed out, “meta-substituted” was modified to “kinked”. See page 2, line 76.

Many thanks to the reviewer for detailed comment.

I hope this revision is satisfactory for your further process. 

Best,

Jin-Hae Chang

Professor

Reviewer 2 Report

in this work, the researchers synthesized two significant series of thermotropic liquid crystal copolymers with different monomer structures and compositions. through changing HBA ratio, the thermal properties, degree of crystallinity, and stability of the liquid crystalline mesophase of the copolymers were evaluated and compared. the superiority properties were obtained and optimized. The molecular dynamics of the two monomers in the two series were evaluated and compared.the research work is innovation and significant, I recommend it to be published.

   the specification of some chart should be more clearly. such as in Figure 4,  for the abscissa, it may be more significant if using mole ratio to substitute mole amount.

Author Response

Dear Editor

This is our response to your comments regarding our paper “Dependence of the Physical Properties and Molecular Dynamics of Thermotropic Liquid Crystalline Copolyesters on p-Hydroxybenzoic Acid Content (polymers-668240) in polymers.

Thank you very much for the reviewer's comments. I have carefully revised the manuscript following the comments of the reviewer.

Response to Reviewer-2 comments

Point-1: Modify mole amount to mole ratio in Figure 4

Answer-1: As the reviewer pointed out, in Figure 4, the mole amount was modified to mole ratio.

Many thanks to the reviewer for detailed comments.

I hope this revision is satisfactory for your further process. 

Best,

Jin-Hae Chang

Professor

Round 2

Reviewer 1 Report

After the modifications introduced by the authors, the manuscript has been significantly improved and it can be acccepted for publication in its present form.